# Avian Orthoreoviruses: A Systematic Review of Their Distribution, Dissemination Patterns, and Genotypic Clustering

**DOI:** 10.3390/v16071056

**Published:** 2024-06-29

**Authors:** Saba Rafique, Farooq Rashid, You Wei, Tingting Zeng, Liji Xie, Zhixun Xie

**Affiliations:** 1SB Diagnostic Laboratory, Sadiq Poultry Pvt. Ltd., Rawalpindi 46000, Pakistan; sabarafique@ymail.com; 2Department of Biotechnology, Guangxi Veterinary Research Institute, Nanning 530001, China; farooq12@mail.ustc.edu.cn (F.R.); weiyou0909@163.com (Y.W.); tingtingzeng1986@163.com (T.Z.); xie3120371@163.com (L.X.); 3Guangxi Key Laboratory of Veterinary Biotechnology, Nanning 530001, China; 4Key Laboratory of China (Guangxi)-ASEAN Cross-Border Animal Disease Prevention and Control, Ministry of Agriculture and Rural Affairs of China, Nanning 530001, China

**Keywords:** avian reovirus, orthoreovirus, reovirus, ARV, genotypic clustering

## Abstract

Avian orthoreviruses have become a global challenge to the poultry industry, causing significant economic impacts on commercial poultry. Avian reoviruses (ARVs) are resistant to heat, proteolytic enzymes, a wide range of pH values, and disinfectants, so keeping chicken farms free of ARV infections is difficult. This review focuses on the global prevalence of ARVs and associated clinical signs and symptoms. The most common signs and symptoms include tenosynovitis/arthritis, malabsorption syndrome, runting–stunting syndrome, and respiratory diseases. Moreover, this review also focused on the characterization of ARVs in genotypic clusters (I–VI) and their relation to tissue tropism or viral distribution. The prevailing strains of ARV in Africa belong to all genotypic clusters (GCs) except for GC VI, whereas all GCs are present in Asia and the Americas. In addition, all ARV strains are associated with or belong to GC I-VI in Europe. Moreover, in Oceania, only GC V and VI are prevalent. This review also showed that, regardless of the genotypic cluster, tenosynovitis/arthritis was the predominant clinical manifestation, indicating its universal occurrence across all clusters. Globally, most avian reovirus infections can be prevented by vaccination against four major strains: S1133, 1733, 2408, and 2177. Nevertheless, these vaccines may not a provide sufficient defense against field isolates. Due to the increase in the number of ARV variants, classical vaccine approaches are being developed depending on the degree of antigenic similarity between the vaccine and field strains, which determines how successful the vaccination will be. Moreover, there is a need to look more closely at the antigenic and pathogenic properties of reported ARV strains. The information acquired will aid in the selection of more effective vaccine strains in combination with biosecurity and farm management methods to prevent ARV infections.

## 1. Introduction

Avian orthoreovirus is a virus that infects birds worldwide and belongs to the genus Orthoreovirus and the family Reoviridae. ARVs have been linked to a range of clinical conditions in chickens, including neurological disorders [1,2], malabsorption syndrome (MAS), stunting syndrome, respiratory/enteric illnesses, pericarditis, myocarditis, hepatitis, immunological depression, and secondary infections caused by various micro-organisms [3,4]. The most frequent problems caused by ARV are arthritis and tenosynovitis, which reduce feed conversion and weight gain in birds and make movement difficult. Flocks infected with ARV had higher mortality rates due to malnutrition and dehydration. However, birds between the ages of 4 and 7 weeks exhibit these clinical signs. ARV poses a major health concern to broiler farmers [3,5]. Chicken carcasses must be removed at slaughter due to the concerning appearance of the damaged hock joints, resulting in significant financial losses [6].

ARV horizontal transmission occurs via viral particle shedding in the digestive tract, as most birds become infected via the fecal–oral route, although transmission via the respiratory tract is also possible. Furthermore, reoviruses can enter through the damaged skin of chicks in the litter, and the virus can spread to hock joints. Vertical transmission to progeny has also been reported [7]. Infections with low pathogenic strains are normally asymptomatic; however, exposure to virulent strains causes tenosynovitis and MAS in immunocompromised birds [8]. ARV is common in all seasons. Meulemanns and Halen [9] discovered that the ARV remained active at temperatures as high as 50 °C, demonstrating the virus’s resilience. The incidence of ARV is lower in summer and fall than in winter and spring [5].

ARV is an icosahedral, nonenveloped virus that is approximately 80 nm in diameter and has a double-capsid structure [10] with a size of 23.42 kb [11]. The ten double-stranded RNA (dsRNA) genome segments are categorized into three size classes, large, medium, and small, and subcategorized into L1, L2, L3, M1, M2, M3, S1, S2, S3, and S4 (Figure 1). These genes encode nonstructural proteins and structural proteins that vary in number [12]. Each segment’s positive strand is identical to its encoded mRNA and contains a type-1 cap at the 5′-end; however, the negative strand contains a pyrophosphate group [13]. The initial seven nucleotides at the 5′-end (GCUUUUU) and the final five nucleotides at the 3′-end (UCAUC) have been conserved in all ARV-positive strands sequenced thus far. This result shows that these sequences may act as target signals for viral transcripts during replication, transcription, and encapsidation [14].

L-class segment: The lambda (λ) proteins are those that the L-class genes encode. The ARV structural protein λA, which the L1 gene encodes, is essential for viral replication and assembly. As the λA protein is highly conserved among many ARV strains, it is a prime target for the development of vaccines and antiviral medications. Although the λB protein is encoded by the L2 gene, its exact sequence is unknown; this protein is thought to be very important for the function of RNA polymerase, which is necessary for viral replication. λC, which is encoded by the L3 gene, is an ARV structural protein that generates turrets that protrude from the fivefold axis of cores and runs from the inner core to the virion’s outer capsid [13].

M-class segment: The micro (μ) refers to proteins that M-class genes encode. The μA is a small subunit of the inner capsid that the M1 gene encodes [15]. μB, the principal translation product of the M2 gene, has an amino terminus that contains a consensus motif for myristoylation [16]. The function of μB in viral replication depends on its myristoylation, and it has been proposed that μB may be involved in the establishment of viral factories [17]. The protein known as μNS is a nonstructural RNA-binding protein that builds up in the viral factories of cells infected with ARV. A more thorough investigation is necessary to verify the idea that μNS is involved in RNA packaging and replication. The whole μNS sequence contains five conserved basic residues that are critical for RNA binding [18].

The S-class segment Sigma (σ) refers to proteins that the S-class genes encode. The minor outer capsid protein σC is a viral cell attachment protein [19]. Antiviral medicines might target σC, which is substantially conserved among ARVs [20]. Several conserved portions of the σC gene, including the 5′ end, 3′ end, and the entire ORF, can be sequenced to classify and genotype avian reoviruses [21].

Despite advances in our knowledge of the biology and variability of these viruses, and the efforts of various groups in the United States [22], Europe [23], Canada [4], and China [24] to identify and type ARV variants, the traditional vaccine strains that are used to immunize commercial flocks, including S1133, 1733, and 2408, have remained unchanged since the 1970s. Due in part to the propensity of RNA viruses for mutation and recombination processes, which might result in variations that are only partially or entirely protected by antibodies produced by traditional vaccination strains, these strains have been shown to be ineffective at managing infection.

The distributions of avian orthoreoviruses, clinical manifestations, and genotypic clustering across continents were the main focus of this study. Additionally, a molecular characterization through a phylogenetic analysis of a subset of reoviruses was conducted using partial S1 gene sequences, specifically the sigma c gene. Sequences were obtained from the National Center for Biotechnology Information (NCBI) and used to trace genotypic clusters worldwide. The evolutionary history was inferred via the maximum likelihood method based on the Tamura–Nei model.

## 2. Clinical Manifestations Based on Genotype

The pathogenicity of ARV is highly variable, and different strains of the virus are linked to illnesses such as tenosynovitis, viral arthritis, and malabsorption syndrome (MAS) [25]. Additionally, these viruses can also be isolated from chicks that show no clinical symptoms. There are many different serotypes and phylogenetic classification schemes; the most popular one is based on the Kant classification [21,26]. Different ARV serotypes and genotypes have been identified under this phylogenetic categorization, and new genotypes, such as genotype 7, continue to arise as a natural consequence of mutation, recombination, and reassortment events during virus replication in Canada [27]. Although the connection between reovirus and tenosynovitis has been demonstrated, whether MAS is caused by reovirus remains unclear [10]. Regardless of the genotypic cluster, tenosynovitis/arthritis was the predominant clinical manifestation, highlighting its universal occurrence across all clusters. GC I and IV exhibit a higher prevalence globally, suggesting a widespread impact. GC IV, in particular, presents with a diverse range of clinical signs, including tenosynovitis/arthritis, runting–stunting syndrome, malabsorption, and numerous cases that remain unidentified. This diversity underscores the complex nature of GC IV and the potential challenges in identifying specific clinical markers. In contrast, GC VI displays a more focused clinical manifestation, with cases primarily characterized by tenosynovitis/arthritis. This distinct clinical profile sets GC VI apart from others, suggesting a more homogenous impact within this genotypic cluster. Notably, within GC I, two isolates also exhibited respiratory signs of infection. This result highlights the variability even within a single genotypic cluster, thus showcasing the importance of considering a broad spectrum of clinical manifestations (Table 1 and Table 2).

## 3. Phylogenetic Analysis Based on the Sigma C Gene

With the use of partial S1 gene characterization methods, ARV strains have been divided into six genotypic groups [21,22,32]. The only available ARV sequences for strains from the USA, Canada, Taiwan, Australia, the Netherlands, Germany, Japan, England, Tunisia, Egypt, Iran, Hungary, Brazil, and China are now available. The gene sequences encoding σC were retrieved from the National Center for Biotechnology Information (NCBI) GenBank (Table 1, Figure 2).

The vaccine isolates are closely related to each other and can be grouped in GC I along with all strains from China, India, and Japan, as well as those isolated from Canada, based on a phylogenetic comparison of the nucleotide sequences of avian orthoreoviruses worldwide [25,32]. These strains differ from isolates from the Netherlands and Australia, which are restricted to GC V and VI, respectively. The isolates from America and Germany are widely dispersed and are divided into five distinct clusters. This result suggests that, in contrast to the Tunisian isolates, which are grouped into a single cluster, these isolates display genetic diversity and are not closely related to one another [25]. In 2019, a new serotype strain, LY383, was isolated, and sequence analysis revealed that the vaccination strains and LY383 are not very comparable. Among all the isolates of ARV of chicken origin, LY383 is grouped into GC V [33].

In California, ARV strains are associated with six different genotypic clusters (GC1 to GC6). The prevalence of ARV, GC I (51.8%) and GC VI (24.7%) were the most common clusters, followed by GC II (12.9%) and GC IV (7.1%), with GC V (2.4%) and GC III (1.2%) having lower rates. Kant described similar outcomes throughout Europe. Few isolates from clusters II and V and the majority of isolates linked to malabsorption syndrome were found in clusters I and IV. The molecular characterization and findings of these compounds have also been reported [27]. The genotypic clusters of ARV isolates in California changed between 2015 and 2018. There was a decrease in the representation of GC I strains and an increase in that of GC VI. Numerous variables, including the use of autogenous vaccinations, may have contributed to this significant change. The ARV genetic clusters that cause disease may be represented differently in the field due to the use of certain GCs as antigens in autogenous vaccines. A valuable method to determine the missing link between strain diversity and pathogenic characteristics is to sequence the whole genomes of ARV strains on a large scale because these extensive data can be used to identify new genetic variants, mutations, and rearrangements in the genetic code.

## 4. Avian Orthoreovirus Distribution Based on Continent

Avian reovirus infections in chickens present a significant global issue to the poultry industry. Thus, determining the distribution of avian orthoreoviruses by continent is essential in order to develop relevant control strategies. Figure 3 details the distribution pertinent to the current review.

### 4.1. Africa

In Egypt, ARV was initially discovered in 1984 [39] and was subsequently detected serologically in several Egyptian governorates [40]. In addition, in Egypt, both vaccinated and nonvaccinated flocks have a high prevalence of ARV. Moreover, 40 nonvaccinated flock members were tested for ARV resistance using RT–PCR, and the σC gene of the virus was sequenced. ARV strains isolated in Egypt do not correspond to GC I, which is the classification of the vaccination strains, but, rather, to GC V [28]. Comparably, seven African isolates were sequenced in a different study on ARV genotypic grouping from global strain collections, and none of them aligned with GC I, although all seven isolates are associated with GC I, II, IV, and V [29]. After being molecularly characterized, the ARV isolates from Tunisia were found to belong to GC I, which also included strains identified from China, England, Japan, and Canada. [41] Previous reports revealed a seroprevalence of 41.0% in grill flocks from Nigeria.

### 4.2. South America

The prevalence of ARV-related illnesses in South America has increased over the past ten years, and this increase has been attributed to pathogenic strains of various lineages [26]. Reports from Brazil showed that arthritis was the cause of the partial or complete removal of many carcasses at slaughterhouses [6,42]. Two previously identified strains, GC II and V, contained novel Brazilian ARV sequences, which were grouped into distinct tree branches. While the GC V sequence of strain BR SC 7001 was comparable to the ARV sequences from Germany, Israel, and the USA, the four ARVs grouped within GC II exhibited greater similarity to isolates from Canada and the USA. The σC sequence variability of ARVs from clinical cases of tenosynovitis in Brazilian poultry flocks was examined [6]. Nevertheless, in addition to the previously published GC II and V, different studies conducted in Brazil later in 2023 revealed additional genotypic clusters, namely, I and III. Subgenotypic clusters I (I vaccine, Ia, Ib), II (IIa, IIb, IIc), and IV (IVa and IVb) might be further separated from this grouping. Four genotypic/subgenotypic clusters comprised Brazilian ARVs: Ib (48.2%), IIb (22.2%), III (3.7%), and V (25.9%) [8].

### 4.3. North America

Based on a nucleotide sequence analysis for the molecular characterization of ARVs, the USA isolates S1133, 1733, 2408, and C08 were shown to be closely related and classified into clusters [21]. Similarly, a phylogenetic study of σC in another study grouped all the isolates from the USA into GC I [32]. Six separate clusters of ARV strains with varying genotypes and degrees of amino acid (aa) sequence similarity were found during 2011 and 2014 by molecular analysis of newly emerging ARV variants in Pennsylvania. Standard ARV vaccine strains and 25 field strains are included in GC I, whereas 38 field strains from GC II, which is very widespread, form three distinct subclusters with varying origins and degrees of aa identity. GC V comprises 27 field strains that differ from the vaccine strains and other reference strains in terms of genotype, making them the next most closely related strains. The seven field strains that had a low aa identity with the reference strains were part of GC III and IV. However, 10 field strains from GC VI are unique and different from all previously released ARV reference strains [22]. Although ARV infection was common (90.5%) among broiler flocks in Ontario, infection was not synonymous with ARV-associated diseases [5].

In contrast to earlier research, a different report revealed genotypic clustering that was divided into four separate subtypes: GC II, IV, V, and VI. No single isolate was categorized as GC III, nor were any of the isolates grouped with the reference vaccination strains. Circulating ARV strains may have developed and diverged greatly from vaccination strains, as evidenced by the isolates having only 53% aa sequence similarity with the American S1133 vaccine strain. Previous research has documented the genotypic clustering of ARV isolates originating from distinct geographical areas [4]. In contrast, GC V was the most prevalent cluster in different studies using grill chickens from Western Canada, followed by GC IV. With only five, four, and three isolates each, GC I, II, and III had the fewest number of isolates overall. There was only one GC VI isolate; however, importantly, four SK sequences that were assigned to that cluster were examined. All five sequences had an identity between 78.2 and 98.6%, and phylogenetic trees classified them as though they were isolated from GC IV. An autogenous vaccination cannot be effective against a challenge virus that is specific to one cluster because of the differences in the genetic composition of the various clusters. GC I and GC VI are the most common, with frequencies of 51.8% and 24.7%, respectively, followed by GC II and GC IV (7–13%) and GC V and GC III (1–2%), which have lower frequencies [27]. Very few isolates belonged to GC II and GC V, while most isolates belonging to MAS were found in GC I and GC IV [27]. Uncertain viruses belonged to GC II, while the majority of tenosynovitis clusters belonged to GC IV [27], indicating that their most common sequences were from GC V, followed by GC IV and GC I [38].

### 4.4. Asia

Tenosynovitis/arthritis syndrome has been more common in the past several years, and many ARV strains have been identified from broilers in China, Korea, the Middle East, and other nations [29,33]. Most isolates from Taiwan are grouped in GC I. Additionally, isolates from Taiwan were also found in GC II and IV. Few MAS isolates are grouped in GC II [21]. There is significant genetic diversity among ARVs in Taiwan. In another report, GC I, II, III, and IV were reported in Taiwan [32].

Up to 95.83% of broiler breeders in the western provinces of Turkey were positive for ARV antibodies [43], while [44] 98.5% of samples positive for ARV were found in Swiss poultry herds. The Iranian province of Tehran had a 98.3% ARV prevalence, demonstrating the need for ARV immunization in chicken flocks. In India, the total prevalence of ARV was 8.67% [45].

A comparative study using the σC sequence revealed that 14 of the 18 isolates were located in GC II, III, and VI, whereas 4/18 isolates were in the same GC I as the vaccination strains. The strains identified in 2017 were categorized into a new genotype known as GC VI, whereas the strains isolated between 2013 and 2016 were mostly found in GC I to GC III [34]. Field strains of ARV belonging to GC V are strongly associated with LY383, an isolate from a Chinese grill flock that received vaccinations. Although these field strains and vaccination strains differ greatly, they have a high level of amino acid similarity with LY383. ARV strains of turkey ancestry created GC II, whereas strains of waterfowl origin developed GC III and IV concurrently [33]. The σC protein exhibited a high level of antigenic homogeneity compared to that of isolates from Chinese chicken origin and commercial vaccines (inactivated vaccines). These isolates, however, were not the same as the isolates from Korean chickens. According to these findings, the inferred amino acid substitution patterns in the σC protein of all the isolates were the same as those of ARVs of chicken origin [35]. According to another report in Japan, GC II and V were found in broiler breeders, whereas GC II and IV were found in layer breeders [30]. A phylogenetic analysis of the σC protein revealed that two field isolates, ARV1IR018 (MAS isolate) and ARV2IR018 (viral arthritis isolate), were clustered in GC IV and II, respectively [36].

### 4.5. Europe

All five genotypic clusters are present in different countries in Europe, including France, Germany, the Netherlands, Spain, Hungary, Romania, and Ukraine [21,23,29,37,46]. GC I, which includes several closely related strains from various locations, including European, Central and South Asian, and North African nations, is the most prevalent in Europe. A different cluster within GC II included European strains originating from the Balkan Peninsula and nearby nations. GC III comprises strains from the same countries that have high sequence identity. Chicken reoviruses in GC IV had the highest genetic diversity and the greatest number of isolates from the worldwide collection. With just a few isolates from Germany and Ukraine sharing high sequence similarities (98%), GC V contained the fewest isolates overall. Additionally, it has been reported that GC V pathogens were detected only in samples taken after 2013. Consequently, GC V is probably less common in Europe than the other clusters [29].

Five distinct genotypic groups were discovered from the categorization of Dutch and German ARVs using σC protein sequencing, with most isolates from Germany falling into GC I. Furthermore, the majority of the German and Dutch MAS isolates under study belong to GC IV. German and Dutch isolates accounted for the majority of GC IV isolates with uncertain cases, and the majority of Dutch isolates with tenosynovitis were classified as GC I. While a small number of MAS isolates from the Netherlands were classified as GC II, isolates from Germany were classified as GC V and III [21]. In contrast, a recent study examined the evolutionary history and genetic diversity of ARV isolates from French grill chickens. Only GC I, which was only tangentially connected to the other clusters and the vaccination strains, contained isolates [23].

Only GC I, III, and V were found in Hungary. With 43 isolates, GC II was the most frequently found cluster in Hungary. Eight strains from Romania and one from Hungary were also discovered by the researchers, and both strains had a significant degree of nucleotide and aa identity similarity with the Hungarian cluster II strains. However, samples from the four nations under study—Hungary, Romania, Russia, and Ukraine—all showed the presence of GC IV. Eleven Hungarian strains that were obtained from the same farm were shown to have strong nucleotide and aa identity similarities with the three Romanian strains. These results demonstrate the notable genetic variation among ARVs in East Central European hens, which may aid in the development of efficient vaccinations and management techniques [37].

### 4.6. Oceania

ARV has been documented within the geographic confines of Australia, although with a lower prevalence in comparison to that in other continents. According to one study, GC V has been exclusively identified [21], while another investigation reported the presence of GC IV and VI [32]. The absence of any other genotypic clusters inside Australia’s boundaries supports the suggestion that ARV transmission in Australia is unlikely to be the result of vaccination and is instead related to migrating birds.

## 5. Vaccine Challenges

Major genetic changes were detected in 1986, with genetic variants from the “conventional vaccine types” of reovirus. This significant change may have been induced by several factors, including the use of autogenous vaccinations. The many ARV genetic clusters producing disease in the field may be changing as a result of the use of certain GCs as antigens in autogenous vaccinations. Although GC I strains of reovirus accounted for most strains in 2016, autogenous vaccines containing isolates of two GC I strains and one GC V variant were produced for use in breeders that provide chickens to the state of California [38,47]. The theory underlying inactivated nonhomologous vaccines is that they provide some protection against the field issue of viral shedding in infected birds, thus allowing strains aside from GC I and GC V to be selected, hence altering the environmental representation of ARVs. However, this justification falls short of explaining why GC IV or GC III were overlooked. Because GC VI is more fit than the other genotypes in the current environment, those genotypes were likely not picked because they were not as fit as the others. Reports on surveillance activities typically do not address how GC detection varies from year to year [21,22,27]. The high variability in sequences causes the reporting of avian orthoreovirus to fluctuate over time. When choosing autogenous vaccine candidates, considering the prevalence of GCs in addition to their antigens is important.

In addition to determining the temporal GC frequencies, the available data permitted the computation of homologies with a reference strain, in this case, the commercial vaccine strain S1133. The benefit is being able to track the variability of each cluster’s variant and determine whether there are any significant changes over time. GC1 has the most homology, and GC1 is the group that includes vaccination strains; however, its average homology was 77%. The average homologies of the remaining GCs to S1133 ranged from 58.5 to 53.1%, which is extremely different from those of viruses found in commercial live and inactivated vaccines (S1133, 1733, and 2408). These findings may explain the ineffectiveness of vaccination in poor defending commercial broilers. Every cluster has remained homologous to S1133 since 2016, according to the homologies reported throughout time [38].

Over the past ten years, pathogenic strains of different lineages have led to an increase in the number of ARV-related infections across North and South America, Europe, Africa, and Asia [4,26,29]. To choose optimal vaccine candidates, thorough and frequent sero- and viro-surveillance is necessary in order to gain an understanding of GC homologies.

## 6. Conclusions

Avian reovirus is now a moving target, similar to the influenza virus, infectious bronchitis virus, and other RNA-based avian pathogens, due to its genetic nature, especially the recombination, genetic drift, and absence of an RNA proofreading mechanism. The incidence of ARVs in the intestines of wild birds was greater than that of ARVs in their excrement [35]. Despite several vaccinations to birds throughout their lives, the chances of infection and reinfection still exist. Overall, several genotypes circulate among the poultry population, and no significant cross-protection has been reported among different genotypes. At least six distinct genotypes were found when ARVs were genotyped utilizing the σC-encoding gene; however, the relationships between genotypes, pathogenic traits, and serotype classifications are still being determined. ARVs exhibit a broad range of tissue tropisms, and virulent ARVs from free-living birds were genetically linked to ARVs from chickens, suggesting that these species might serve as reservoirs for the spread of ARVs within poultry farms and could become a moving target that needs to be monitored through regular sero- and viro-surveillance.

In addition, GC I and IV have a higher prevalence globally, indicating a widespread impact. The clinical signs of GC IV are particularly diverse and include tenosynovitis/arthritis, runting–stunting syndrome, malabsorption, and unidentified cases. This diversity underscores the complexity of GC IV and the challenges in identifying specific clinical markers. In contrast, GC VI has a more focused clinical manifestation, primarily characterized by tenosynovitis/arthritis. This characteristic sets GC VI apart from others, suggesting a more homogeneous impact within this genotypic cluster. Moreover, GC I isolates also exhibit respiratory signs of infection, thereby highlighting the variability even within a single genotypic cluster. This result emphasizes the importance of considering a broad spectrum of clinical manifestations.

A phylogenetic analysis revealed that the vaccine isolates were closely related to each other and categorized into GC I. This grouping includes strains from China, India, Japan, and Canada, in addition to the vaccine isolates. However, these isolates are distinct from those originating from the Netherlands and Australia, which are classified as GC V and VI, respectively. Furthermore, isolates from Germany and the United States are widely distributed and form five distinct clusters based on their genetic similarity. This result suggests a diverse genetic landscape for avian orthoreoviruses in these regions.

This review further indicated, based on continent prevalence, that there is no exact correlation between ARV genotypes and geographic location. Furthermore, point mutation accumulation and reassortment processes play a critical role in the evolution of ARVs. The nonspecific geographic distribution of all six ARV genotyping cluster groups indicated that vaccine formulations containing appropriate antigens from all six genotypes are necessary for the successful prevention of viral-induced arthritis/tenosynovitis. In vaccinated breeders or broiler flocks, novel variant strains lead to vaccine breakthroughs. The antigenic and pathogenic characteristics of a few of the ARV strains that have been identified will be further examined. In an effort to control ARV infections, the information gathered here will facilitate more effective vaccination strain selection with biosecurity and farm management practices.

## Figures and Tables

**Figure 1 viruses-16-01056-f001:**
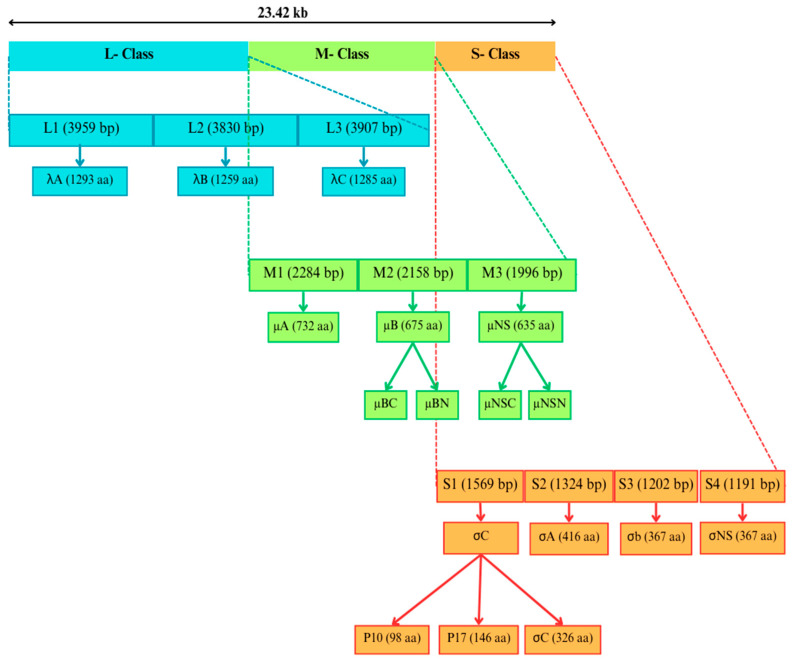
An overview of the avian orthoreovirus genetic structure.

**Figure 2 viruses-16-01056-f002:**
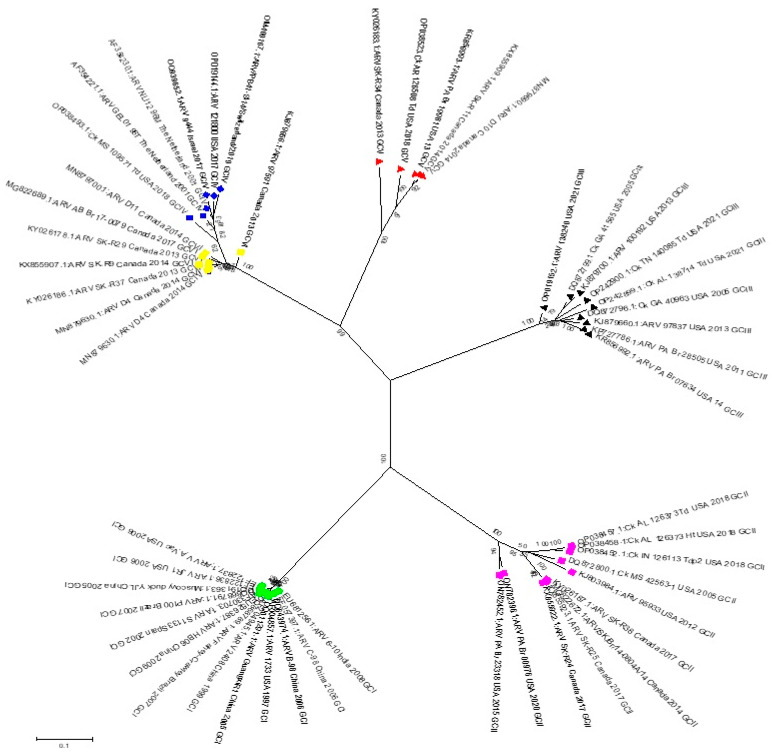
Phylogenetic tree of avian orthoreovirus strains based on the nucleotide sequence of the S1 gene (sigma c). Phylogenetic trees were constructed using MEGA-6 software by the maximum likelihood method with 1000 bootstrap replicates. The symbols are as follows: green circle, GC I; purple square, GC II; black triangle, GCIII; blue rhombus, GC IV; red triangle, GC V; and yellow square, GC VI.

**Figure 3 viruses-16-01056-f003:**
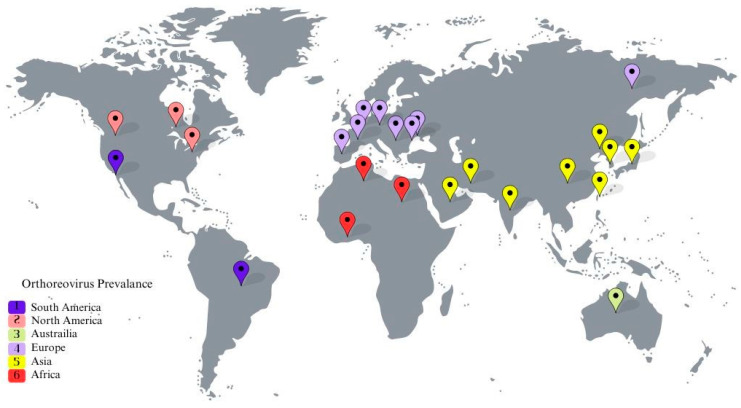
Prevalence of ARV by continent. The global distribution of ARV is illustrated on a world map with different pins marking each continent.

**Table 1 viruses-16-01056-t001:** Summary of avian orthoreovirus prevalence, reference strains, clinical signs, and genotypic clusters based on continent.

Continent	Country	Reference Strains	Clinical Symptoms	Genotypic Clusters						Reference
Africa	Tunisia	TU430	Viral arthritis, and malabsorption syndrome	I						[25]
		TU105B6								
		TU5								
		TU97.2								
	Egypt	Chicken/Egypt/Gharbia/1-20/2020	Stunted growth and enlarged lemon-shaped proventriculus with reduced gizzard size					V		[28]
		Chicken/Egypt/Gharbia/2-20/2020								
		D2572/3/2/14EGVAR2	Tenosynovitis/arthritis and runting–stunting syndrome		II	III	IV	V		[29]
		D2248/1/2/13EG								
		D2095/1/4/12EG								
		D2929/2/1/15EG								
Asia	Japan	ARV Bro NGN20 7-1 2b	Viral arthritis/tenosynovitis, malabsorption syndrome, and runting–stunting syndrome		II		IV	V		[30]
		ARV Bro GF20 3-7 b								
		ARV Lay NIGT20 30B2b								
		ARV Lay NIGT20 40J1 a								
		ARV Bro NGN20 7-1 b								
		ARV Bro GF20 4-1 a								
		JP/Tottori/2016	N/A	I					VI	[31]
		JP/Nagasaki/2017								
		OS161	Malabsorption	I						[32]
	China	ARV LY383 China 2016	Arthritis/tenosynovitis, runting–stunting syndrome, hepatitis, myocarditis, MAS, and central nervous system disease					V		[33]
		HeN130728	Foot and joint swelling, hemorrhage, foot scab, paralysis and lameness	I	II	III			VI	[34]
		LN160607-1								
		GX150816								
		SD150806								
		JS170705-1								
	South Korea	A15-48/Wild bird/Korea 2015	Viral arthritis/tenosynovitis, MAS, runting–stunting syndrome, and respiratory diseases	I						[35]
		A18-205/Wild bird/Korea2018								
	Iran	ARV2IR018	Viral arthritis and MAS		II		IV			[36]
		ARV1IR019								
	Taiwan	750505	Tenosynovitis, respiratory, and MAS	I	II		IV			[21]
		T6								
		916								
		1017-1								
		T6	Respiratory disease, viral arthritis, and MAS	I	II	III	IV			[32]
		601G								
		918								
		916								
Europe	France	11-12523	Poor growth rates, lameness, ruptures of gastrocnemicus tendons, and nonuniform bodyweights	I						[23]
		11-17268								
		12-1167								
	Germany	GEL06 97M	Tenosynovitis, and MAS	I	II	III	IV	V		[21]
		GEL13 98M								
		GEL01 96T								
		GEL15 00M								
	The Netherland	NLI03 92T	Tenosynovitis, and MAS	I	II		IV			[21]
		NLI20 98M								
		NLA13 96T								
	Hungary	HUN131	Lameness, runting–stunting syndrome, and uneven growth rate	I	II	III	IV	V		[37]
		HUN392								
		HUN290								
		HUN142								
		HUN385								
	Romania	ROM6	Runting–stunting syndrome		II		IV			
		ROM11								
	Ukraine	UKR1	Mortality				IV			
	Russia	RUS1	Runting–stunting syndrome				IV			
North America	Canada	SK R38	Unilateral or bilateral inflammation of tendons, ruffled feathers, lameness, splayed legs, and reluctance to get up and walk		II		IV	V	VI	[4]
		RAM-1								
		05682/12								
		NLI12 96M								
	Pennsylvania	Reo/PA/Broiler/05273a/14	Lesions of pericarditis, swelling, edema, and hemorrhages in the tendons and tendon sheath		II	III	IV	V	VI	[22]
		Reo/PA/Broiler/07634/14								
		Reo/PA/Broiler/30857/11								
		Reo/PA/Broiler/07209a/13								
		Reo/PA/Broiler/03476/12								
	Canada	17-0160-Broiler-AB-2017	Unilateral lameness, subcutaneous hemorrhage, rupture of tendon, and secondary bacterial infection	I	II	III	IV	V	VI	[27]
		14-0041-Broiler-SK-2014								
		16-0711-Broiler-BC-2016								
		12-1009-Broiler-AB-2012								
		15-0157-Broiler-BC-2015								
		17-0025-Broiler-AB-2017								
	California	MK247039	Swollen hock joints, lameness, stunting, and lack of uniformity	I	II	III	IV	V	VI	[38]
		MK247050								
		MK246988								
		MK247008								
		MK247040								
		MK247049								
South America	Brazil	Reo/BR_Sc_6996	Synovial membranes hyperplasia, inflammation, hemorrhages, fibrin deposition, and necrosis of muscle fibers		II			V		[6]
		Reo/BR_Sc_7001								
		BR-3118	Tenosynovitis, and MAS	I	II	III		V		[8]
		BR/2290								
		BR-5881								
		BR/3292								
Oceania	Australia	SOM-4	Unclear					V		[21]
		RAM-1	Healthy							
		RAM-1	Healthy					V	VI	[32]
		SOM-4	Viral arthritis							

**Table 2 viruses-16-01056-t002:** Clinical manifestations of avian orthoreoviruses according to genotype.

Clusters	TS	RS	TS/RS	MAS	RES	Other	ND	Healthy	Continent	References
GC I	38	2	0	21	2	12	8	2	Africa, America, Asia, Europe, Middle East	[8,21,25,29,32]
GC II	9	7	1	2	0	18	6	0	Africa, America, Asia, Europe, Middle East	[8,21,29,32]
GC III	2	1	0	3	0	9	2	0	Africa, America, Asia, Europe, Middle East	[8,21,29,32]
GC IV	17	10	1	16	0	29	12	2	Africa, America, Asia, Europe, Middle East	[21,29,32]
GC V	7	3	0	3	0	1	2	2	Africa, America, Asia, Europe, Middle East, Australia	[8,21,29,32]
GC VI	46	0	0	0	0	0	0	0	America, Asia, Australia	[22,27,31,32,34,38]

TS: tenosynovitis/arthritis, RS: runting–stunting syndrome, TS/RS: tenosynovitis/arthritis or runting–stunting syndrome, MAS: malabsorption syndrome, RES: respiratory disease, ND: not defined.

## Data Availability

Data sharing is not applicable to this article.

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
