# Peer review of "Avian Orthoreoviruses: A Systematic Review of Their Distribution, Dissemination Patterns, and Genotypic Clustering"

_viruses, 2024, doi:10.3390/v16071056_

Round 1

Reviewer 1 Report

Comments and Suggestions for Authors

This MS described a systemic review of avian orthorevirus. I believed a lot of  useful information has been included and presented by the author in this MS. However, too much fragmented information without a comprehensive organization still disturbed the readers. Thus, it was suggested that the author needs a thorough rephrasing and reconstructing the body of this MS.

Specific comments were as follows:

1. Please combine Table 1 and Table 2. 

2. line 155: Please explain why the isolates from Germany and the United States are the most widely distributed?

3. lines 165-166: The format of this sentence was incorrect.

4. lines 171-172: Please explain what is the advantage of the whole genome sequence since most people useed the analysis based on the sigma C?

5. lines 199-200: Please use past tense.

6. lines 208-209: Grammar was wrong.

7. line 210: Plese check your writing. Actually I suggested you need the English editing again.

8. lines 271-274: Very lousy construction of sentences. Please rephrase it.

9. lines 333-335: The format of this sentence was unacceptable.

10. lines 334-335: The grammar needs to be improved. 

11. Could you include some figures to detail Part 4. Avian orthoreovirus distribution continent-wise for each country. All readers were buried by too long and tedious descriptions from the authors. We need clear and distinct statements. 

Comments on the Quality of English Language

Please send your MS for the further Englisg editing. 

Author Response

Reviewer 1

This MS described a systemic review of avian orthorevirus. I believed a lot of useful information has been included and presented by the author in this MS. However, too much fragmented information without a comprehensive organization still disturbed the readers. Thus, it was suggested that the author needs a thorough rephrasing and reconstructing the body of this MS.

Specific comments were as follows:

  1. Please combine Table 1 and Table 2.

Reply: Thank you for your comment, we have now changed the heading of table 2, and thus its overall context is changed.

  1. line 155: Please explain why the isolates from Germany and the United States are the most widely distributed?

Reply: The isolates from America and Germany are widely dispersed and are divided into five distinct clusters. This suggests that, in contrast to the Tunisian isolates, which are grouped into a single cluster, these isolates display genetic diversity and are not closely related to one another. Please refer to line No. 160-164 (Highlighted in grey in the text)

  1. lines 165-166: The format of this sentence was incorrect.

Reply: Thank you for your comment. We have now rearranged this sentence. Please refer to line No. 172-173 (Highlighted in grey color in the text).

  1. lines 171-172: Please explain what is the advantage of the whole genome sequence since most people useed the analysis based on the sigma C?

Reply: Thank you for this comment. We have now given the reason in the text. Please refer to line No. 178-181 (highlighted in grey in the text).

  1. lines 199-200: Please use past tense.

Reply: Thank you for this comment. We have now used the past tense. Please refer to line No. 212 (highlighted in grey).

  1. lines 208-209: Grammar was wrong.

Reply: Thank you for the comment. We have now corrected the grammar. Please refer to line No. 219 (highlighted in grey)

  1. line 210: Plese check your writing. Actually I suggested you need the English editing again.

Reply: Thank you for the comment. We have now rephrased the sentence to make it more clear. Please refer to line No.221-222 (highlighted in grey).

  1. lines 271-274: Very lousy construction of sentences. Please rephrase it.

Reply: Thank you for your comment. We have now rephrased the sentence. Please refer to line No. 278-281 (highlighted in grey)

  1. lines 333-335: The format of this sentence was unacceptable.

Reply: Thank you for your comment. We have now rephrased the sentence. Please refer to line No. 340-342 (highlighted in grey).

  1. lines 334-335: The grammar needs to be improved.

Reply: Thank you for your comment. We have now rephrased the sentence. Please refer to line No. 340-342 (highlighted in grey).

  1. Could you include some figures to detail Part 4. Avian orthoreovirus distribution continent-wise for each country. All readers were buried by too long and tedious descriptions from the authors. We need clear and distinct statements.

Reply: Thank you for this comment. We have now incorporated new figure (Figure 3).

Reviewer 2 Report

Comments and Suggestions for Authors

The paper "Avian Orthoreovirus: A Systemic Review of their distribution, dissemination patterns, and genotypic clustering" is a sorely needed review on what has been published in genotypic cluster. However, there are some areas that need to be addressed. I have the following comments. 

General comments

- Neurological signs can also be caused by ARV (van den zande et al 2007). Please, update the paper where necessary. 

- Please, comment on the paper the different classifications between laboratories and countries exist for ARV and how confusing can it be for researchers and field veterinarians to have a classification in place. 

- There is an account of a possible "Genotype 7" (Palomino-Tapia et al 2022).

Specific comments

- Line 75-77. This is in reference to a 2004 paper. This is not recent. Please, modify the text. 

-  Line 80. "Similar to protein...". This is confusing. Please, rewrite. 

- Lines93-96. Please, specify which part of Sigma C protein is substantially conserved. 

- Lines 236-248. Please, rewrite for clarity. Try adding a citation after each sentence to track for findings. 

- Line 319. "Major genetic changes occurred starting in 1986...". Please, change for "changes detected in 1986."

-Lines 356-357. Add a citation. 

Comments on the Quality of English Language

English is good. Some areas seem to have been added / moved and need to be checked for grammar. 

Author Response

Reviewer 2

The paper "Avian Orthoreovirus: A Systemic Review of their distribution, dissemination patterns, and genotypic clustering" is a sorely needed review on what has been published in genotypic cluster. However, there are some areas that need to be addressed. I have the following comments.

General comments

- Neurological signs can also be caused by ARV (van den zande et al 2007). Please, update the paper where necessary.

Reply: Thank you for this comment. We have now updated this sentence and added this reference as well. Please refer to line No. 37-38 (highlighted in yellow).

- Please, comment on the paper the different classifications between laboratories and countries exist for ARV and how confusing can it be for researchers and field veterinarians to have a classification in place.

-There is an account of a possible "Genotype 7" (Palomino-Tapia et al 2022).

Reply: Thank you for the comment, and we agree to your point as well. However, there are many different serotypes and phylogeny classification schemes; the most popular one is based on the Kant classification. Different ARV serotypes and genotypes have been identified under this phylogenetic categorization, and new genotypes such as genotype 7 continue to arise as a natural consequence of mutation, recombination, and re-assortment events during virus replication in Canada. Please refer to line No. 115-119 (highlighted in yellow).

Specific comments

- Line 75-77. This is in reference to a 2004 paper. This is not recent. Please, modify the text.

Reply: Thank you for the comment. Since we think that this was not important, therefore, we have now removed this sentence.

-  Line 80. "Similar to protein...". This is confusing. Please, rewrite.

Reply: Thank you for the comment. We have now rephrased it. Please refer to line No. 77 (highlighted in yellow).

- Lines93-96. Please, specify which part of Sigma C protein is substantially conserved.

Reply: We have provided the information in the text. Please refer to line No. 93-94 (highlighted in yellow).

- Lines 236-248. Please, rewrite for clarity. Try adding a citation after each sentence to track for findings.

Reply: Thank you for your comment. We have now rephrased the sentences and added citations. Please refer to line No. 242-248 (highlighted in yellow).

- Line 319. "Major genetic changes occurred starting in 1986...". Please, change for "changes detected in 1986."

Reply: Thank you for your comment. We have now changed it according to the instructions. Please refer to line No. 319 (highlighted in yellow).

-Lines 356-357. Add a citation. 

Reply: Thank you for your comment. We have now added a citation. Please refer to line No. 358 (highlighted in yellow).

Round 2

Reviewer 2 Report

Comments and Suggestions for Authors

Please, review your citations. Citation 27 should refer to a 2022 publication in Avian diseases. 

Lines 327-329 - Rewrite for clarity. 

Comments on the Quality of English Language

English is good. Some areas seem to have been added / moved and need to be checked for grammar.

Some sentences, like the ones in 327-329, seem to not have been checked for grammar once corrected. 

Author Response

Comments and Suggestions for Authors

Please, review your citations. Citation 27 should refer to a 2022 publication in Avian diseases. 

Reply: Thank you for your comment. We have thoroughly investigated this citation. Moreover, we have also now added the citation from Avian Diseases, reference number 47, line number 328 and 494.

Lines 327-329 - Rewrite for clarity. 

Reply: Thank you for your comment. We have now rephrased this sentence to make it more clear. Please refer to line no. 328-331 (highlighted in green).

Comments on the Quality of English Language

English is good. Some areas seem to have been added / moved and need to be checked for grammar.

Some sentences, like the ones in 327-329, seem to not have been checked for grammar once corrected. 

Reply: Thank you for your comment. We have now rephrased this sentence to make it more clear. Please refer to line no. 328-331 (highlighted in green).